# Connexin43 Region 266–283, via Src Inhibition, Reduces Neural Progenitor Cell Proliferation Promoted by EGF and FGF-2 and Increases Astrocytic Differentiation

**DOI:** 10.3390/ijms21228852

**Published:** 2020-11-23

**Authors:** Rocío Talaverón, Esperanza R. Matarredona, Alejandro Herrera, José M. Medina, Arantxa Tabernero

**Affiliations:** 1Instituto de Neurociencias de Castilla y León (INCYL), Universidad de Salamanca, 37007 Salamanca, Spain; rtalaveron@usal.es (R.T.); medina@usal.es (J.M.M.); 2Departamento de Fisiología, Universidad de Sevilla, 41012 Sevilla, Spain; matarredona@us.es (E.R.M.); alejandrohd92@hotmail.com (A.H.)

**Keywords:** neural progenitor cells, neurons, astrocytes, ß-catenin, glioma stem cells, connexin, neural precursors, Cx43

## Abstract

Neural progenitor cells (NPCs) are self-renewing cells that give rise to the major cells in the nervous system and are considered to be the possible cell of origin of glioblastoma. The gap junction protein connexin43 (Cx43) is expressed by NPCs, exerting channel-dependent and -independent roles. We focused on one property of Cx43—its ability to inhibit Src, a key protein in brain development and oncogenesis. Because Src inhibition is carried out by the sequence 266–283 of the intracellular C terminus in Cx43, we used a cell-penetrating peptide containing this sequence, TAT-Cx43_266–283_, to explore its effects on postnatal subventricular zone NPCs. Our results show that TAT-Cx43_266–283_ inhibited Src activity and reduced NPC proliferation and survival promoted by epidermal growth factor (EGF) and fibroblast growth factor-2 (FGF-2). In differentiation conditions, TAT-Cx43_266–283_ increased astrocyte differentiation at the expense of neuronal differentiation, which coincided with a reduction in Src activity and β-catenin expression. We propose that Cx43, through the region 266–283, reduces Src activity, leading to disruption of EGF and FGF-2 signaling and to down-regulation of β-catenin with effects on proliferation and differentiation. Our data indicate that the inhibition of Src might contribute to the complex role of Cx43 in NPCs and open new opportunities for further research in gliomagenesis.

## 1. Introduction

Neural progenitor cells (NPCs) are a heterogeneous population of self-renewing and multipotent cells that give rise to the three major cell types in the developing central nervous system: neurons, astrocytes, and oligodendrocytes [1,2]. NPCs persist in the postnatal and adult brain in two specific regions: the subventricular zone (SVZ) of the lateral ventricles and the dentate gyrus of the hippocampus [3]. In addition to their physiological role, there is strong evidence showing that NPCs from the SVZ could be the cell of origin of glioblastoma, the most common and devastating primary brain tumor [4,5,6,7,8,9]. Multiple mechanisms and molecules have been reported to be involved in the control of NPC survival, proliferation, migration, and differentiation towards neurons, astrocytes, and oligodendrocytes. Connexins (Cxs), a family of proteins that form gap junctions (GJs) and hemichannels, are expressed in NPCs of the embryonic, postnatal, and adult ventricular zone and SVZ [10,11,12,13]. Evidence of functional hemichannels and GJ-mediated communication among SVZ cells in vivo and in vitro has been described [12,14]. Furthermore, several reports have indicated that intercellular communication through GJs plays an important role in the maintenance of NPC proliferation [15], survival [13,16], and pluripotency [17].

Among the 21 Cx isoforms, Cx43 has been revealed to be crucial during neural development with important roles in NPC migration and fate (reviewed in [18,19]). Indeed, in Cx43-null brains, the thickness of the cortical plate is found to be reduced, and this is accompanied by an impairment in cortical lamination [20,21]. The defect in neural development found in Cx43-deficient mice might be caused by a defective migration [22,23] and/or by the induction of premature neuronal differentiation [21,24]. Interestingly, some of the Cx43-mediated effects on NPC migration, proliferation, and differentiation are exerted through GJ-independent mechanisms [21,22,23,24]. Thus, the knockdown of Cx43 in NPCs, but not the inhibition of GJs, increases neuronal differentiation at the expense of astrocyte differentiation through the up-regulation of ß-catenin [24], a transcription factor required for neuronal differentiation [25,26,27]. Intriguingly, in NPCs, this negative modulation of neuronal differentiation is carried out by the intracellular C terminus of Cx43 (Cx43-CT) [21].

Cx43-CT interacts with a plethora of signaling partners [28], such as Src [29], a non-receptor protein tyrosine kinase that regulates a wide range of cellular events related to cell survival, proliferation, differentiation, and migration [30,31,32]. In addition to its well-known role in cancer and particularly in glioblastoma [33], Src is abundantly expressed in the developing nervous system, playing a key role in neural development [34,35,36,37]. Cx43 inhibits the activity of Src [38] by recruiting Src together with its endogenous inhibitors, CSK and PTEN [39]. This effect takes place in the region expanded from amino acid 266 to 283, within the intracellular Cx43-CT [40]. Indeed, the cell-penetrating peptide TAT-Cx43_266–283_ recapitulates the inhibition of Src promoted by Cx43 in astrocytes and in a broad spectrum of glioma models, including human glioma stem cells, in vitro and in vivo [39,40,41,42]. In fact, these studies have shown the efficacy of TAT-Cx43_266–283_ as a therapy against glioblastoma in preclinical models [39,40,41,42].

Because Cx43, and specifically Cx43-CT, is crucial for NPC fate, and Src is a key molecule in neural development, in this study, we hypothesized that Src could be a mediator in some of the GJ-independent effects of Cx43 in NPCs. Therefore, we investigated the role of the region of Cx43-CT, responsible for Src regulation, by using TAT-Cx43_266–283_ in postnatal SVZ-derived NPCs in proliferation and differentiation conditions. We found that the region of Cx43 (266–283), depending on the cellular context, could affect NPC proliferation, survival, and differentiation and that these effects were mediated by Src inhibition.

## 2. Results

### 2.1. Effects of TAT-Cx43_266–283_ in NPCs in Proliferation Conditions

To investigate the effect of the sequence 266–283 in Cx43 in NPCs, we fused it to the penetrating peptide TAT (TAT-Cx43_266–283_). First, we evaluated the effect of TAT-Cx43_266–283_ on the size of the neurospheres formed when neurosphere-derived NPCs were mechanically dissociated and cultured with 20 ng/mL epidermal growth factor (EGF) and 10 ng/mL fibroblast growth factor-2 FGF-2, i.e., in proliferation conditions. Cells cultured in these conditions are referred to as NPCs in proliferation conditions (PC NPCs) as illustrated in Figure 1A.

Our results show that TAT-Cx43_266–283_ reduced the growth of neurospheres in a dose-dependent way. Thus, the size of neurospheres was smaller in cultures treated for 72 h with 25 or 50 µM TAT-Cx43_266–283_, but not with 10 µM, compared to cultures treated with TAT sequence alone (Figure 2A). Due to the strong effect observed with 50 µM TAT-Cx43_266–283_ in PC NPCs (Figure 2A), the subsequent experiments were carried out with 25 µM. It should be mentioned that no significant differences were found in neurosphere size among PC NPCs untreated (control) or treated with 25 μM TAT or 25 μM control peptide, TAT-Cx43_274–291_, whereas NPCs treated with TAT Cx43_266–283_ at the same concentration showed a significant reduction in neurosphere size compared to all these conditions (Figure 2B).

In order to know whether the impairment in the growth of neurospheres observed in PC NPCs treated with TAT-Cx43_266–283_ was due to a decrease in cell proliferation, to an increase in cell death, or to both mechanisms, we evaluated cell proliferation and apoptosis in PC NPCs by Ki67 and caspase-3 immunostaining, respectively. As shown in Figure 3A, NPC treatment with 25 µM TAT-Cx43_266–283_ for 72 h significantly reduced the percentage of proliferating cells from 84 ± 3% to 64 ± 3%. In addition, TAT-Cx43_266–283_ treatment of PC NPCs also produced a significant increase in the percentage of apoptotic cells (Figure 3B). Therefore, both a reduction in mitotic activity and an increase in apoptosis may account for the observed TAT-Cx43_266–283_-induced decrease in neurosphere size.

Because Src is involved in the signaling pathway evoked by EGF and FGF-2 [43,44], and TAT-Cx43_266–283_ reduces Src activity in astrocytes and in glioma stem cells in vitro and in vivo [42,45], we aimed to confirm whether Src activity is also inhibited by TAT-Cx43_266–283_ in PC NPCs. To do so, we analyzed c-Src activity by measuring the levels of c-Src phosphorylated at Tyr-416 (Y416 c-Src), the active form of this tyrosine kinase [46,47]. Our results show that the incubation of PC NPCs with 25 µM TAT Cx43_266–283_ for 72 h produced a marked reduction (of about 80%) in Src activity (Figure 3C and Figure A1A). Altogether, our results suggest that TAT-Cx43_266–283_ might reduce NPC proliferation and survival by mechanisms dependent on Src activity inhibition. To support this notion, we performed experiments with the Src inhibitor Dasatinib (1 µM), and we obtained a similar effect on neurosphere size than that obtained with 25 μM TAT-Cx43_266–283_ (Figure 4A). In addition, Dasatinib treatment of PC NPCs significantly reduced the percentage of proliferating cells and increased the percentage of apoptotic cells (Figure 4B,C). Together, these data indicate that TAT-Cx43_266–283_ reduced NPC proliferation and survival promoted by EGF and FGF-2 through the inhibition of Src activity.

### 2.2. Effects of TAT-Cx43_266–283_ in NPCs in Differentiation Conditions

Some studies have reported that in NPCs, Cx43 acts as a negative modulator of neuronal differentiation [21,24]. Therefore, our next goal was to evaluate the effect of TAT-Cx43_266–283_ in NPCs growing in differentiation conditions (DC NPCs). To do so, neurosphere-derived NPCs were mechanically dissociated and seeded on an adhesive substrate with reduced levels of EGF and FGF-2, as illustrated in Figure 1A. TAT-Cx43_266–283_, TAT-Cx43_274–291_, or TAT (all of them at 25 μM) were added to the differentiation medium, and cultures were maintained for 96 h after which differentiation to neurons and astrocytes was evaluated. We observed that differentiation towards neurons was inhibited by TAT-Cx43_266–283_. Thus, the number of DCX-positive cells when NPCs were cultured with TAT-Cx43_266–283_ was significantly reduced when compared to NPCs cultured with TAT or with the control peptide TAT-Cx43_274–291_ (Figure 5A).

Concomitantly, TAT-Cx43_266–283_ promoted NPC differentiation to astrocytes, as demonstrated by the strong increase in the number of GFAP immunoreactive cells (Figure 5B). The effect on astrocyte differentiation was also specific to the 266–283 amino acid sequence since treatment with TAT-Cx43_274–291_ did not produce a significant change in astrocytic differentiation (Figure 5B). The reduction in DCX expression and the increase in GFAP expression promoted by TAT-Cx43_266–283_ were also corroborated by western blotting (Figure 5C,D and Figure A1B). The effect of TAT-Cx43_266–283_ treatment on Cx43 expression in DC NPCs was also evaluated, and no significant changes were identified in the number of Cx43-positive cells between TAT- and TAT-Cx43_266–283_-treated cultures (Figure A2).

Next, we investigated the mechanism by which TAT-Cx43_266–283_ affects NPC differentiation. Our results show that TAT-Cx43_266–283_ inhibited Src activity in DC NPCs (Figure 6A and Figure A1C). To corroborate that the inhibition in Src activity could account for the effects found with TAT-Cx43_266–283_ in NPC differentiation, we analyzed NPC fate in DC NPCs treated with the Src inhibitor Dasatinib. Treatment with 1 µM Dasatinib also induced a marked decrease in neuronal differentiation in DC NPCs (Figure 6B), as it happened with TAT-Cx43_266–283_. However, in contrast to the effect observed with TAT-Cx43_266–283_, NPC differentiation towards astrocytes was not significantly affected by Dasatinib treatment (Figure 6C). It should be mentioned that, in addition to inhibiting Src, Dasatinib also inhibits other protein tyrosine kinases, such as BCR-ABL, c-KIT, PDGFR, and ephrinA2 [48]. Therefore, it could be speculated that some of these signaling pathways might be required for NPC differentiation to astrocytes. In agreement with the relevance of some of these pathways for astrocytes, Dasatinib has been shown to reduce astrocyte viability and migration [42].

Previous reports have demonstrated that the knockdown of Cx43 increases NPC differentiation towards neurons by up-regulating the expression of ß-catenin [24]. Because the activity of Src elevates the expression of ß-catenin [49], we explored the effect of TAT-Cx43_266–283_ on the levels of β-catenin in DC NPCs. As shown in Figure 6D, 25 µM TAT-Cx43_266–283_ induced a strong decrease in β-catenin expression in DC NPCs (Figure 6D and Figure A1D), which coincided with the reduction in c-Src activity (Figure 6A and Figure A1C). These results suggest that the Cx43 amino acid sequence 266–283, by inhibiting Src activity, led to a decrease in ß-catenin levels, which reduced neuronal differentiation at the expense of astrocytic differentiation.

## 3. Discussion

The role of Cx43 in NPC survival, proliferation, and differentiation has been extensively studied. Among the different functions of Cx43, intercellular communication through GJs has been clearly demonstrated to play an important role in maintaining NPCs in a proliferative state [13,15,16,17], while the Cx43-CT plays a more relevant role in NPC migration and differentiation [21,23]. In this study, we focused on one specific property of Cx43—its ability to inhibit the activity of Src, which is located in the Cx43-CT, specifically in sequence 266–283. Our study unveils that this region of Cx43 (266–283), depending on the cellular context, can affect NPC proliferation, survival, and differentiation and that these effects are mediated by Src inhibition.

It is well established that NPC survival and proliferation as neurospheres in vitro require the presence of mitogens, such as EGF and FGF-2 [50]. Src is involved in the signaling pathway evoked by EGF and FGF-2 [43,44]. Therefore, it is not unexpected that TAT-Cx43_266–283_, by inhibiting Src activity, impairs NPC survival and proliferation when they are cultured in the presence of these mitogens, causing the reduction in the size of neurospheres. Most studies show that gap junctional communication between NPCs is required for their survival and proliferation [15,16,17]. Interestingly, in this study, we found that Cx43 through the region 266–283 inhibited Src activity with the subsequent reduction in NPC survival and proliferation promoted by EGF and FGF-2 mitogens. These different channel-dependent and -independent functions of Cx43 could contribute to the maintenance of a correct number of NPCs, avoiding their aberrant overgrowth, which could contribute to malignant glioma formation. In fact, an increase in EGF or FGF-2 levels within the SVZ niche promotes an over-cell proliferation and hyperplasia [51,52,53]. In addition, EGF receptor amplification in NPCs gives rise to the development of malignant gliomas [4]. Because TAT-Cx43_266–283_ inhibits NPC growth promoted by EGF and FGF-2 via Src inhibition, this peptide might be used to target NPCs carrying mutations that lead to glioblastoma development. Further studies are needed to address this interesting possibility. In fact, Src is well recognized as a crucial target for glioblastoma [33]. Although so far, the results from clinical trials with the Src inhibitor, Dasatinib, have been discouraging, most evidence suggests that more specific Src inhibitors with a higher ability to cross the blood–brain barrier are required to improve glioblastoma therapy [54,55].

Previous studies have shown that down-regulation of Cx43 drives NPC differentiation toward neurons, decreasing astrocyte differentiation [24]. This effect depends on the Cx43-CT, which is crucial for the signaling mechanisms, preventing premature neuronal differentiation during embryonic brain development being, therefore, a negative modulator of neuronal differentiation [21]. In this study, we found that the Cx43 region 266–283 was sufficient to reduce neuronal differentiation in favor of astrocyte differentiation. Interestingly, the effect of the down-regulation of Cx43 on favoring NPC differentiation toward neurons at the expense of astrocyte differentiation has been reported to be mediated by the up-regulation of β-catenin expression [24]. Although this study clearly shows that Cx43 regulated β-catenin expression, the molecular mechanism underlying this effect is not totally known. It has been proposed that the interaction between Cx43 and β-catenin sequestrates β-catenin, preventing its translocation to the nucleus and, consequently, inhibiting its transcriptional activity [56]. This interaction explains how Cx43 regulates β-catenin activity; however, it does not explain the effect of Cx43 on β-catenin expression. Importantly, our study reveals that the Cx43 region 266–283 reduced β-catenin expression. It should be mentioned that β-catenin interaction with Cx43 occurs at Cx residues 259–275, 282–295, and 302–319 [57]; therefore, it is unexpected that Cx43 region 266–283 reduced β-catenin expression by its sequestration. Because Src activity increases β-catenin expression [49], and we found in this study that Cx43 region 266–283 reduced Src activity and β-catenin expression in DC NPCs, it could be suggested that Src is involved in the effect of Cx43 on β-catenin expression. Consequently, we propose that Cx43 in NPCs, through the region 266–283, reduces Src activity with the subsequent down-regulation of β-catenin expression, which favors astrocyte differentiation at the expense of neuronal differentiation. In conclusion, our results show that the region 266–283 in Cx43 inhibits Src in NPCs, which can have different roles, depending on the cellular context. Thus, when the concentration of mitogens is high, the region 266–283 in Cx43 might interfere in the increase in NPC survival and proliferation promoted by these growth factors, while upon a reduction in the concentration of mitogens, it might favor astrocyte differentiation to the detriment of neuronal differentiation. Together these data strengthen the interest of Src in NPC biology and show that the inhibition of Src promoted by Cx43 might contribute to the complex role of Cx43 in NPC survival, proliferation, and differentiation, which might also be relevant for brain tumor development.

## 4. Materials and Methods

### 4.1. Animals

Albino Wistar rats were obtained from the animal facility of the Universidad de Sevilla and Universidad de Salamanca (Spain). The animal procedures were approved by the ethics committee of Universidad de Sevilla and by the bioethics committee of the Universidad de Salamanca and Junta de Castilla y León (Spain) and were carried out in accordance with the guidelines of the European Union (2010/63/EU) and Spanish law (R.D. 53/2013 BOE 34/11370-420, 2013) for the use and care of laboratory animals.

### 4.2. NPC Culture in Proliferation and Differentiation Conditions

NPC cultures were obtained from 7-day postnatal (P7) rats of either sex as previously described [58]. Four P7 rats were used for every independent culture. Briefly, coronal slices of the rat brain were obtained, and the lateral walls of the lateral ventricles containing the SVZ were removed (Figure 1A) and enzymatically dissociated with 1 mg/mL trypsin (Invitrogen, Thermo Fisher Scientific, Waltham, MA, USA) at 37 °C for 15 min. The tissue was then centrifuged at 150 g for 5 min, rinsed in Dulbecco’s modified Eagle’s medium/F12 medium 1:1 (DF-12; Invitrogen), and centrifuged again in the same conditions. Then, the cells were resuspended in DF-12 containing 0.7 mg/mL ovomucoid (Sigma-Aldrich, St Louis, MO, USA) and mechanically disaggregated with a fire-polished Pasteur pipette. The dissociated cells were centrifuged and resuspended in DF-12 containing B-27 supplement minus vitamin A, 2 mM Glutamax^®^, 100 unit/mL penicillin, 100 µg/mL streptomycin, and 0.25 µg/mL amphotericin B, all from Invitrogen, supplemented with 20 ng/mL epidermal growth factor (EGF; PeproTech, Rocky Hill, NJ, USA) and 10 ng/mL basic fibroblast growth factor (FGF-2; Millipore, Temecula, CA, USA). The cells were maintained at 37 °C in an atmosphere of 95% air/5% CO_2_ and with 90–95% humidity. After 1–2 days, floating cell aggregates of NPCs known as neurospheres were formed. Cultures maintained in these conditions are referred to as NPCs in proliferation conditions (PC NPCs, Figure 1A,B). PC NPCs were subcultured every 72−96 h by centrifugation, mechanical dissociation, and resuspension with fresh medium. All the experiments were performed with cells between passages 2 and 6.

Differentiation of NPCs was performed as previously reported [59]. Briefly, neurospheres, obtained as described above, were mechanically dissociated and seeded on poly-L-ornithine-treated 12-mm diameter coverslips (for immunohistochemistry) or on poly-L-lysine-treated T25 flasks (for Western blot experiments) in DF-12 with 1% fetal calf serum (FCS; Gibco, Life Technologies, Madrid, Spain) at a density of 10,000 cells/cm^2^. Both adhesive substrates were purchased from Sigma Aldrich. After 4 h, the medium was replaced by DF-12 containing B-27 supplement with vitamin A (Invitrogen), 2 mM Glutamax^®^, 100 units/mL penicillin, 100 µg/mL streptomycin, 0.25 µg/mL amphotericin B, and reduced concentrations of EGF (0.8 ng/mL) and FGF-2 (0.4 ng/mL). Cultures maintained in these conditions are referred to as NPCs in differentiation conditions (DC NPCs, Figure 1A,C). After 96 h, coverslips were fixed with 4% paraformaldehyde in 0.1 M phosphate buffer for 10 min, after which they were kept in phosphate buffer saline (PBS):glycerol (1:1) until immunostaining.

### 4.3. Cell Treatments

The synthetic peptides (>85% pure) used in this study were obtained from GenScript (Piscataway, NJ, USA). YGRKKRRQRRR was used as the TAT sequence, which is responsible for the cell penetration of the peptides [40]. The TAT-Cx43_266–283_ sequence was TAT-AYFNGCSSPTAPLSPMSP, and the TAT-Cx43_274–291_ sequence was TAT-PTAPLSPMSPGYKLVTG. Previous studies have shown that TAT-Cx43_274–291_ lacks the ability to inhibit the oncogenic activity of Src [41] because it does not recruit Src with its inhibitors [39] as TAT-Cx43_266–283_ does. Therefore, TAT-Cx43_274–291_ is used as a peptide control for the effects of TAT-Cx43_266–283_ related to Src inhibition [41]. These peptides were added to the culture medium at 10, 25, or 50 µM for 72 h in PC NPCs or at 25 µM for 96 h in DC NPCs.

The Src inhibitor, Dasatinib, and its vehicle, dimethyl sulfoxide (DMSO), were purchased from Sigma Aldrich. Dasatinib (1 μM) or DMSO (1 μL/mL) were added to the culture medium for 72 h in PC NPCs or for 96 h in DC NPCs.

The peptides, Dasatinib or DMSO, were added to the culture medium when neurosphere-derived cells were seeded, and the treatments were maintained for the indicated times (Figure 1A).

### 4.4. Analysis of Neurosphere Size, Proliferation, and Apoptosis in PC NPCs

Experiments were performed in 24-well plates seeded with neurosphere-derived cells at a density of 10,000 viable cells/cm^2^ in proliferation conditions. Wells were treated at the time of seeding with different concentrations (10, 25, and 50 µM) of TAT, TAT-Cx43_266–283_, or TAT-Cx43_274–291_ or with 1 μM Dasatinib or its vehicle (DMSO, 1 μL/mL). Seventy-two hours after seeding, photographs were captured from neurospheres of each well using a Leica EC3 camera coupled to a phase-contrast microscope (Leica DMIL-LED) with a 10× objective (eight photographs of random fields per well). The diameter of the neurospheres was measured with ImageJ (NIH).

In another set of experiments, proliferation and apoptosis were evaluated in PC NPCs by Ki67 and caspase-3 immunostaining, respectively. For that purpose, after 72 h of treatment, neurospheres were mechanically dissociated and seeded on poly-L-ornithine-treated coverslips with DF-12 containing 1% FCS to allow adhesion to the coverslips. After two hours, coverslips were fixed with 4% paraformaldehyde in 0.1 M phosphate buffer for 10 min and kept in PBS:glycerol (1:1) until immunostaining.

### 4.5. Immunocytochemistry

Cells on coverslips were blocked with 2.5% bovine serum albumin (Sigma-Aldrich) in PBS (blocking solution) for 30 min and incubated with primary antibody for 12 h at 4 °C. After rinsing, cells were incubated with the secondary antibody for 60 min at 25 °C. Cells were counterstained with 4′-6′-diamidino-2-phenylindole (DAPI, Sigma-Aldrich, 0.1 µg/mL) for 10 min, washed again, and mounted on slides with an n-propyl-gallate solution (Sigma-Aldrich, 0.1 M) prepared in glycerol:PBS 9:1. The primary antibodies used were Ki67 (mouse monoclonal, Sigma Aldrich; Ref. P6834; 1:200), caspase-3 (rabbit polyclonal, Cell Signaling, Danvers, MA, USA; Ref. 9662; 1:400), doublecortin (DCX, a marker for neurons, mouse monoclonal, Santa Cruz Biotechnology, Dallas, Texas, USA; Ref. 271390; 1:100), glial fibrillary acidic protein (GFAP, a marker for astrocytes, mouse monoclonal, Sigma Aldrich; Ref. G3893; 1:1400), and Cx43 (mouse monoclonal, BD Biosciences, Ref. 610062 1: 50). The secondary antibodies used were anti-mouse IgG labeled with TRITC or with FITC (Jackson Immuno Research, West Grove, PA, USA, Ref. 715-025-151; Ref. 715-095-151; 1:200), anti-mouse IgG labeled with Alexa Fluor 488, and anti-rabbit IgG labeled with Alexa Fluor 594 (Life Technologies, Thermo Fisher Scientific, Waltham, USA; Ref. A28175; Ref. R37115; 1:1000). Antibodies were prepared in blocking solution. Fluorescent images of the cells were captured at 20× magnification using a camera DP73 (Olympus) coupled to an epifluorescence microscope Olympus BX61 or with a camera DC100 (Leica Microsystems, Wetzlar, Germany) coupled to a Nikon Eclipse TS2000 microscope. The omission of primary antibodies resulted in the absence of detectable staining in all cases.

Six random fields were captured per coverslip, and at least three coverslips seeded with neurosphere-derived cells from different cultures were analyzed in each experimental condition. Counting of Ki67-, caspase-3-, DCX-, GFAP-, or Cx43-positive cells was performed in the images and expressed as a percentage of the total number of cells identified by DAPI staining.

### 4.6. Western Blot Analysis

Western blotting was performed as previously described [38]. Briefly, equivalent amounts of proteins (40 µg per lane) were separated on NuPAGE Novex Bis-Tris (4–12%) midigels (Life Technologies). The proteins were transblotted using an iBlot dry blotting system (Life Technologies). After blocking, the membranes were incubated overnight at 4 °C with the primary antibodies against rabbit Y416 Src (Cell Signaling; Ref. 2101; 1:200), rabbit total c-Src (Cell Signaling; Ref. 2108; 1:500), guinea pig DCX (Chemicon International, Merck Millipore, Madrid Spain; Ref. AB5910; 1:1000), mouse GFAP (Sigma; Ref. G3893; 1:500), or mouse β catenin (BD Bioscience, Madrid, Spain; Ref. 610153; 1:500). The antibodies against mouse glyceraldehyde phosphate dehydrogenase (GAPDH, Ambion, Thermo Fisher Scientific; Ref. AM4300; 1:15,000) and mouse L-ribosomal protein (RPL, Santa Cruz Biotechnology Ref. Sc-100830; 1:500) were used as a loading control. After extensive washing, the membranes were incubated with 1:5000 peroxidase-conjugated anti-rabbit IgG, anti-mouse IgG antibody (Santa Cruz Biotechnology, Dallas, TX, USA; Refs. sc-2030 and sc-2005), or anti-guinea pig IgG antibody (Sigma Aldrich; Ref. A7289) in Tween-Tris-buffered saline and developed with a chemiluminescent substrate (Western Blotting Luminol Reagent; Santa Cruz Biotechnology) in a MicroChemi imaging system (Bioimaging Systems).

Quantification of Western blots was performed using the ImageJ program following the Gel Analysis method. The parameter analyzed was the relative density. The amounts of GAPDH or RPL recovered in each sample served as the loading control, and the values for each protein were normalized to their corresponding GAPDH or RPL.

### 4.7. Statistical Analysis

Results are expressed as the mean ± SEM of at least three independent experiments. Statistical analyses were carried out in Sigma Plot 11 (Systat Software). Student’s *t*-test for parametric or Mann–Whitney Rank Sum test for non-parametric data was used when two groups were compared. For more than two groups, one-way analysis of variance (ANOVA) was used, followed by Holm–Sidak.

## 5. Patents

ES2526109B1. Peptide and pharmaceutical composition for cancer treatment.

## Figures and Tables

**Figure 1 ijms-21-08852-f001:**
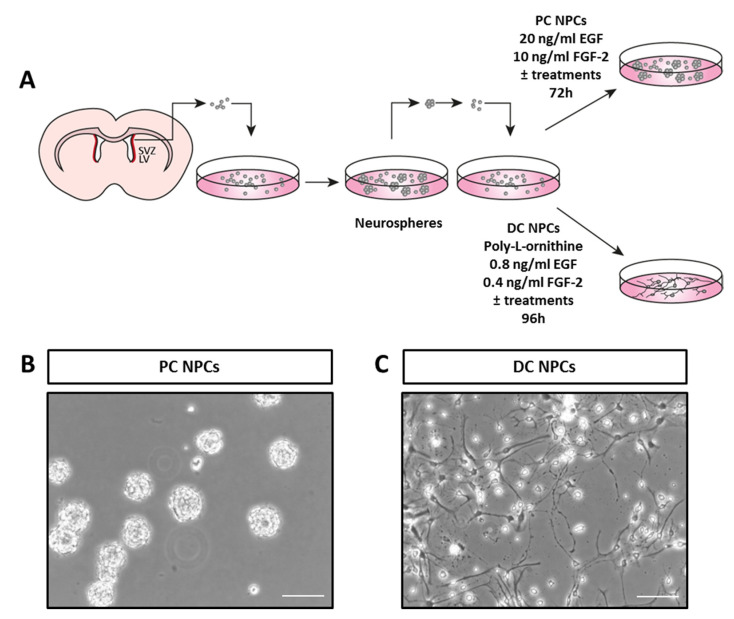
Neural progenitor cells of the postnatal subventricular zone in proliferation and differentiation conditions. (**A**). Schematic drawing of the design of the protocol used in this study. The subventricular zones (SVZ, in red) adjacent to the lateral ventricles (LV) were dissected out from coronal slices obtained from 7-day postnatal rat brains, and neural progenitor cells (NPCs) contained in this region were cultured in the form of neurospheres, as described in Methods. Neurosphere-derived NPCs were exposed to different treatments in proliferation conditions (PC) or in differentiation conditions (DC). (**B**). Phase-contrast photomicrograph showing neurospheres or floating cell aggregates of NPCs in proliferation conditions (PC NPCs) without treatments. (**C**). Phase-contrast photomicrograph showing NPCs grown in differentiation conditions (DC NPCs) without treatments. Bar: 100 µm. EGF (epidermal growth factor), FGF-2 (fibroblast growth factor).

**Figure 2 ijms-21-08852-f002:**
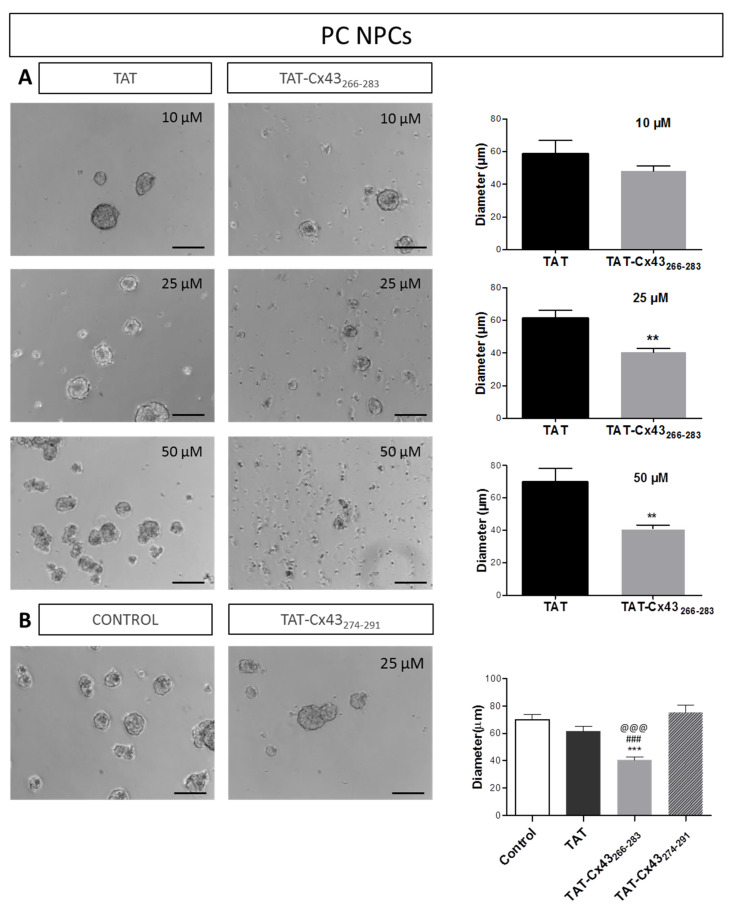
Effect of TAT-Cx43_266–283_ on neurosphere size in proliferation conditions. (**A**) Phase-contrast images of neural progenitor cells grown in proliferation conditions (PC NPCs) after treatment with 10, 25, or 50 μM of TAT or TAT-Cx43_266–283_ for 72 h. Bar: 100 µm. Quantification of neurosphere diameter after 72 h treatment with 10–50 µM of TAT or TAT-Cx43_266–283_. Data are the mean ± SEM of *n* = 6–7 (** *p* < 0.01, Mann–Whitney Rank Sum Test). (**B**) Phase-contrast images of PC NPCs untreated (control) or treated with 25 μM control peptide, TAT-Cx43_274–291_, for 72 h. Bar: 100 µm. Quantification of neurosphere diameter in untreated cultures (control) or cultures treated with 25 µM TAT, 25 μM TAT-Cx43_266–283_, or 25 μM TAT-Cx43_274–291_ for 72 h. Data are the mean ± SEM, *n* = 12 (control), *n* = 9 (TAT), *n* = 7 (TAT-Cx43_266–283_), *n* = 4 (TAT-Cx43_274–291_) (*** *p* < 0.001 vs. control; ^###^
*p* < 0.001 vs. TAT; ^@@@^
*p* < 0.001 vs. TAT-Cx43_274–291_, ANOVA, followed by Holm–Sidak).

**Figure 3 ijms-21-08852-f003:**
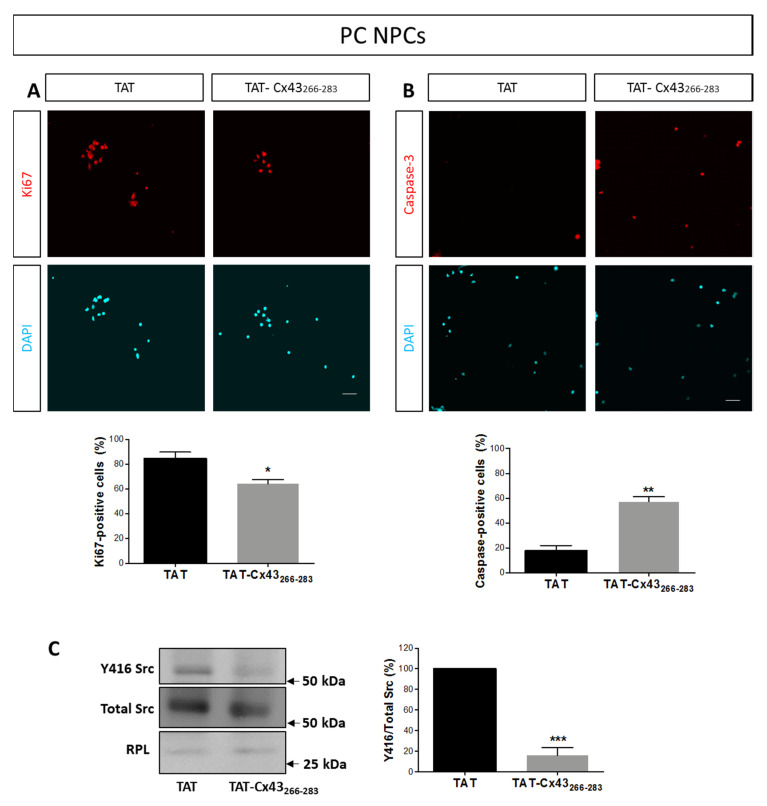
Effect of TAT-Cx43_266–283_ on neural progenitor cell proliferation, apoptosis, and Src activity in proliferation conditions. (**A**,**B**) Representative epifluorescence images and quantifications of Ki67 (**A**, in red) or caspase-3 (**B**, in red) immunostaining in cultures of neural progenitor cells in proliferation conditions (PC NPCs) treated with 25 μM TAT or TAT-Cx43_266–283_ for 72 h. Cell nuclei of the same fields were visualized by DAPI staining (turquoise). Bar: 75 µm. Results are expressed as percentage of Ki67-positive cells (in **A**) or caspase-3-positive cells (in **B**) and are the mean ± SEM (*n* = 3; * *p* < 0.05; ** *p* < 0.01; Student’s *t*-test). (**C**) Western blotting showing Y416 Src and total Src in PC NPCs treated with 25 µM TAT or TAT-Cx43_266–283_ for 72 h. Quantification of Y416 Src/Src ratio expressed as a percentage with respect to TAT. Results are expressed as mean ± SEM (*n* = 4; *** *p* < 0.001; Student’s *t*-test).

**Figure 4 ijms-21-08852-f004:**
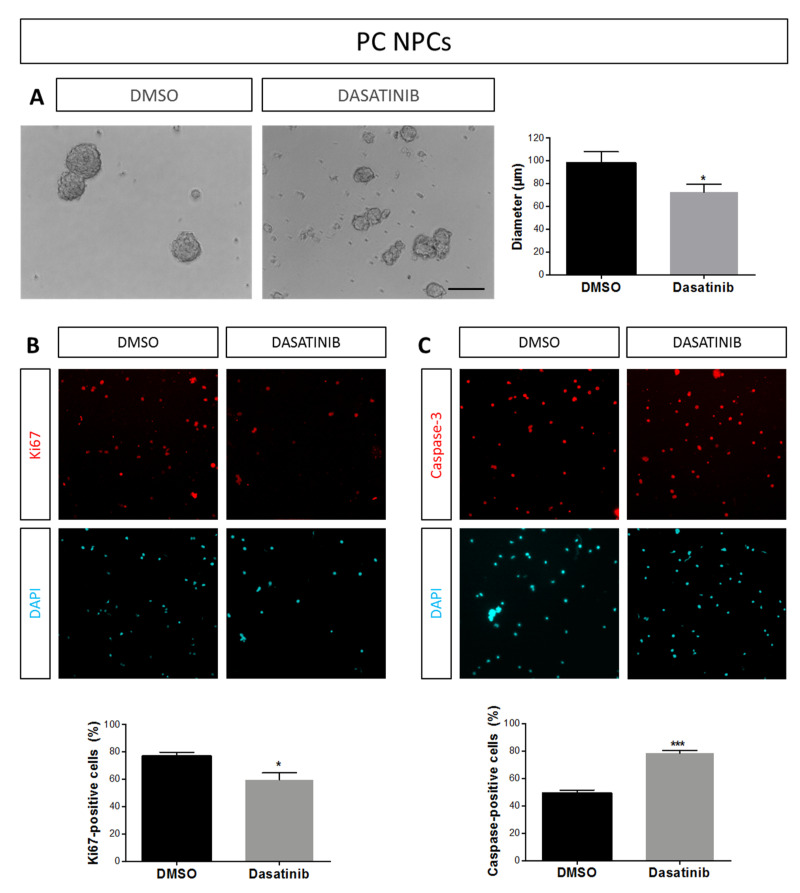
Effect of Dasatinib on neurosphere size, neural progenitor cell proliferation, and apoptosis in proliferation conditions. (**A**) Phase-contrast images of neural progenitor cells in proliferation conditions (PC NPCs) after 72 h treatment with dimethyl sulfoxide (DMSO) or with 1 μM Dasatinib. Bar: 100 µm. Quantification of neurosphere diameter after 72 h treatment with DMSO or with 1 μM Dasatinib. Data are the mean ± SEM (*n* = 8; * *p* < 0.05 compared to DMSO; Mann–Whitney Rank Sum Test). (**B**,**C**) Representative epifluorescence images and quantifications of Ki67 (**B**, in red) or caspase-3 (**C**, in red) immunostaining in PC NPCs treated with DMSO or with 1 μM Dasatinib for 72 h. Cell nuclei of the same fields were identified by DAPI staining (turquoise). Bar: 75 µm. Results are expressed as percentage of Ki67-positive cells or caspase-3-positive cells and are the mean ± SEM (*n* = 3; * *p* < 0.05, *** *p* < 0.001; Student’s *t*-test).

**Figure 5 ijms-21-08852-f005:**
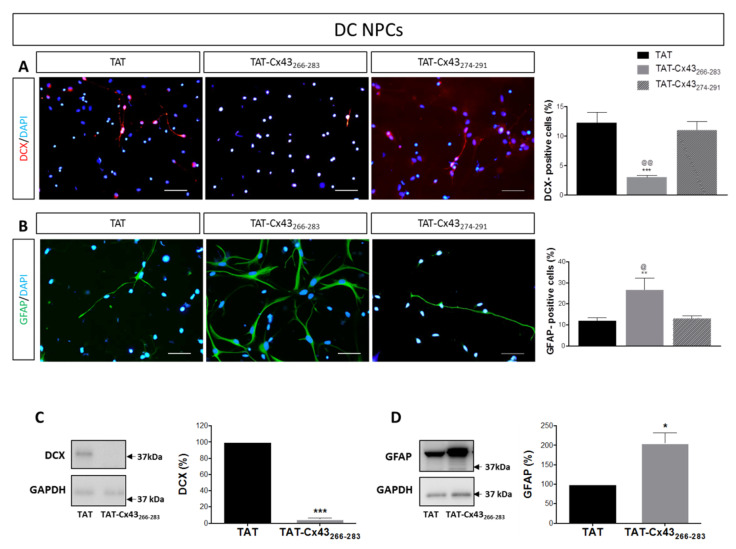
Effect of TAT-Cx43_266–283_ on neural progenitor cells in differentiation conditions. (**A**,**B**) Representative images showing doublecortin (DCX) immunoreactivity (red) and glial fibrillary acidic protein (GFAP) immunoreactivity (green) in neural progenitor cells grown in differentiation conditions (DC NPCs) treated with 25 µM TAT, TAT-Cx43_266–283_, or TAT-Cx43_274–291_ for 96 h. Cell nuclei are labeled with DAPI (blue). Bar: 50 µm. Bar graphs show the percentage of DCX-positive and GFAP-positive cells after 96 h of treatment in each experimental condition. Results are the mean ± SEM (*n*= 9 TAT, *n* = 4 TAT-Cx43_266–283_, *n* = 3 TAT-Cx43_274–291_; *** *p* < 0.001, ** *p* < 0.01 vs. TAT; ^@@^
*p* < 0.01, ^@^
*p* < 0.05 vs. TAT-Cx43_274–291_; ANOVA, followed by Hom Sidak). (**C**,**D**) Total DCX and GFAP levels were analyzed by Western blot in DC NPCs treated with 25 µM TAT or TAT-Cx43_266–283_ for 96 h. Results were normalized against GAPDH and are expressed as percentage with respect to TAT (mean ± SEM, *n* = 4; * *p* < 0.05; *** *p* < 0.001; Student’s *t*-test).

**Figure 6 ijms-21-08852-f006:**
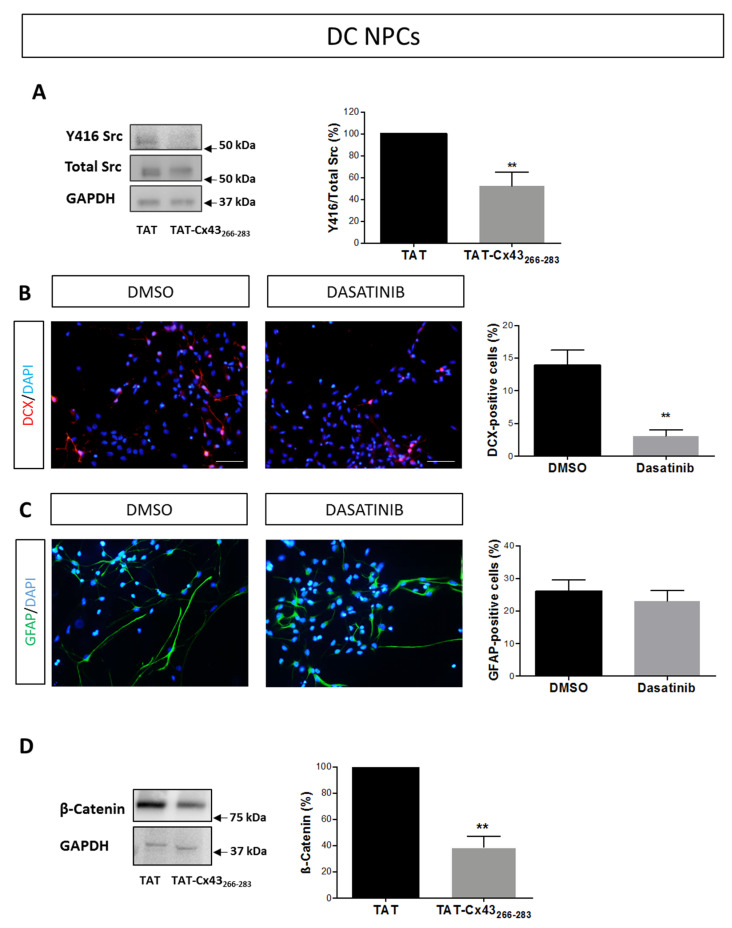
Involvement of Src and β-catenin in the effects of TAT-Cx43_266–283_ on neural progenitor cell differentiation. (**A**) Western blots showing Y416 Src and total Src in neural progenitor cells treated with 25 µM of TAT or TAT-Cx43_266–283_ in differentiation conditions for 96 h. Quantification of Y416 Src/Src ratio expressed as a percentage of TAT. Results are the mean ± SEM (*n* = 5; ** *p* < 0.01; Student’s *t*-test). (**B**) Representative images showing doublecortin (DCX) immunoreactivity (red) in neural progenitor cells in differentiation conditions (DC NPCs) after 96 h of treatment with DMSO or with 1 µM Dasatinib. Cell nuclei were labeled with DAPI (blue). Bar: 50 µm. Percentage of DCX-positive cells in both experimental conditions. Results are shown as mean ± SEM (*n* = 5; ** *p* < 0.01; Student’s *t*-test). (**C**) Representative images showing glial fibrillary acidic protein (GFAP) immunoreactivity (green) in DC NPCs after 96 h of treatment with DMSO or with 1 µM Dasatinib. Cell nuclei are labeled with DAPI (blue). Bar: 50 µm. Percentage of GFAP-positive cells in both experimental conditions. Results are shown as mean ± SEM (*n*= 6; non-significant differences; Student’s *t*-test). (**D**) β-catenin levels were analyzed by western blotting in DC NPCs treated with 25 µM of TAT or TAT-Cx43_266–283_ for 96 h. The bar graph shows the ratio of β-catenin/GAPDH in percentage with respect to TAT (mean ± SEM; *n* = 3; ** *p* < 0.01; Student’s *t*-test).

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
