# Peer review of "Connexin43 Region 266–283, via Src Inhibition, Reduces Neural Progenitor Cell Proliferation Promoted by EGF and FGF-2 and Increases Astrocytic Differentiation"

_ijms, 2020, doi:10.3390/ijms21228852_

Round 1

Reviewer 1 Report

In this study the authors have focused on the inhibition of Src by the sequence 266-283 present on the carboxi terminal of Cx43 situated on neural progenitor cells. They showed that this region of Cx43 can affect NPC proliferation, survival and diferentiation mediated by Src inhibition.

The authors did a great job on the graphical abstract and figures of the manuscript. The manuscript is well written and concise with clear conclusions.

Minor things to fix:

  1. Line 12: the word correspondence is repeated.
  2. Figures 3, 5, 6, and S1: on the western blot figures the arrows are not pointing to the bands, please fix them. 
  3. Legends of figures 3 and 5: Lines 129 and 208: instead of "as percentage of TAT" should said in percentaje respect to TAT. Because you are not showing percentage of TAT, you are showing percentage of protein expression for each of the TAT treatments.

Author Response

Reviewer 1

Comments and Suggestions for Authors

In this study the authors have focused on the inhibition of Src by the sequence 266-283 present on the carboxi terminal of Cx43 situated on neural progenitor cells. They showed that this region of Cx43 can affect NPC proliferation, survival and diferentiation mediated by Src inhibition.

The authors did a great job on the graphical abstract and figures of the manuscript. The manuscript is well written and concise with clear conclusions.

We thank the reviewer for the positive and encouraging comments and for stressing the clarity and concision of this study. The reviewer's suggestions have improved the article. Below, you can find our point-by-point responses to all of the comments.

Minor things to fix:

  1. Line 12: the word correspondence is repeated.

The reviewer is correct. This has been amended in the revised version of the manuscript.

  1. Figures 3, 5, 6, and S1: on the western blot figures the arrows are not pointing to the bands, please fix them. 

The arrows are not pointing to the bands because they are pointing to the position of the molecular weight markers used in the electrophoresis. In the revised version of the manuscript we have enclosed full Western blot images in which the position of these markers is more clearly shown (please see, new Figure 1S).

  1. Legends of figures 3 and 5: Lines 129 and 208: instead of "as percentage of TAT" should said in percentaje respect to TAT. Because you are not showing percentage of TAT, you are showing percentage of protein expression for each of the TAT treatments.

The reviewer is correct. This has been amended in the revised version of the manuscript. Thank you for your suggestion.

Reviewer 2 Report

This article is of excellent merit as it describes the mechanism of how the inhibition of Cx43 inhibited Src activity would reduced NPC proliferation and survival. In addition, this work showed that In differentiation conditions, -Cx43  inhibition increased astrocyte differentiation at expense of neuronal differentiation, coupled with a reduction in Src activity and β-catenin expression.

major concern

The work is sound however, it would be of extreme importance that the authors assess the gap junction and hemichannels alterations and their integrity via assessing their proteins.

The data contains a lot of Western blotting, the authors should provide the whole blots as well as the replicates (n =3)

Author Response

Reviewer 2

This article is of excellent merit as it describes the mechanism of how the inhibition of Cx43 inhibited Src activity would reduced NPC proliferation and survival. In addition, this work showed that In differentiation conditions, -Cx43  inhibition increased astrocyte differentiation at expense of neuronal differentiation, coupled with a reduction in Src activity and β-catenin expression.

We thank the reviewer for the supportive and positive comments highlighting the merit of this study.The reviewer's suggestions have improved the article. Below, you can find our point-by-point responses to all of the comments.

major concern

The work is sound however, it would be of extreme importance that the authors assess the gap junction and hemichannels alterations and their integrity via assessing their proteins.

In line with the reviewer’s suggestion, new experiments have been carried out to asses Cx43 by immunocytochemistry in DC NPCs treated with TAT-Cx43266-283 (please see new supplementary Figure 2S and the Results section). As you can see, the percentage of cells expressing Cx43 is not significantly modified by TAT-Cx43266-283 treatment.

The data contains a lot of Western blotting, the authors should provide the whole blots as well as the replicates (n =3)

In accordance with the reviewer’s suggestion, the new supplementary Figure 1S has been modified to include the full Western blot images as well as the replicates (n=3, 4 or 5) that were previously shown. We would like to point out that blots were routinely cut horizontally into smaller pieces according to the molecular weight, and each piece was incubated with a different antibody. This procedure allows the evaluation of different proteins in the same samples improving the results because the proteins analysed, including the loading control, belong to the same loaded sample.

Reviewer 3 Report

Talaveron et al. convincingly demonstrate that Cx43 (region 266-283) reduces Src activity, leading to disruption of EGF / FGF-2 signaling, which results in suppression of proliferation and differential effects on neuronal/astrocytic differentiation in vitro. With regard to previous publications by the same group (Oncogene 2010, Oncotarget 2016, Cell Death Disease 2014) the novelty is marginal. However, the experimental design is well-selected, suitable statistics were performed, adequate control conditions were selected, and the manuscript is well-written.

I have only two minor comments:

  • Regarding Fig 4: a targeted Src knockdown would corroborate the findings that are reported upon Dasatinib treatment
  • In the discussion section, current status and challenges of targeting Src by Dasatinib in glioblastoma should be briefly discussed (i.e. Neuro Oncol. 2015 Jul; 17(7): 910–911.), otherwise the reader lacks a clear relation to glioma/gliomagenesis/CNS tumor oncogenesis

Author Response

Talaveron et al. convincingly demonstrate that Cx43 (region 266-283) reduces Src activity, leading to disruption of EGF / FGF-2 signaling, which results in suppression of proliferation and differential effects on neuronal/astrocytic differentiation in vitro. With regard to previous publications by the same group (Oncogene 2010, Oncotarget 2016, Cell Death Disease 2014) the novelty is marginal. However, the experimental design is well-selected, suitable statistics were performed, adequate control conditions were selected, and the manuscript is well-written.

We thank the reviewer for the positive and encouraging comments highlighting the design and quality of the manuscript. We would like to stress that the novelty of this study is that we have focused on neural progenitors cells.The reviewer's suggestions have improved the article. Below, you can find our point-by-point responses to all of the comments.

I have only two minor comments:

Regarding Fig 4: a targeted Src knockdown would corroborate the findings that are reported upon Dasatinib treatment.

We have previously shown that Dasatinib inhibits Src and affects proteins targeted by Src, such as Id1, Sox-2 or N- and E-Cadherin in glioma stem cells. Please see the attached Figure 6 from Gangoso et al., 2014 (PMID: 24457967).

In the discussion section, current status and challenges of targeting Src by Dasatinib in glioblastoma should be briefly discussed (i.e. Neuro Oncol. 2015 Jul; 17(7): 910–911.), otherwise the reader lacks a clear relation to glioma/gliomagenesis/CNS tumor oncogenesis.

In accordance with the reviewer’s suggestion, the current status and challenges of targeting Src by Dasatinib in gliomas (Neuro Oncol. 2015 Jul; 17(7): 910–911), as well as the need for new Src inhibitors (Cirotti et al, 2020; Cancers 2020, 12, 1558; doi:10.3390/cancers12061558) have been discussed in the new version of the manuscript.

Round 2

Reviewer 2 Report

Thry answered my questions

Author Response

We thank the reviewer for indicating that we have answered his/her questions